# Hospital Environmental Factors That Influence Peripheral Oxygen Saturation Measurements: A Cross-Sectional Study

**DOI:** 10.3390/healthcare12090940

**Published:** 2024-05-03

**Authors:** Candelaria de la Merced Díaz-González, Cristina Pérez-Bello, Milagros De la Rosa-Hormiga, Juan José González-Henríquez, María de las Mercedes Reyes-Noha

**Affiliations:** 1Department of Nursing, Faculty of Health Sciences, University of Las Palmas de Gran Canaria, 35016 Las Palmas de Gran Canaria, Canary Islands, Spain; cristina.perez@ulpgc.es (C.P.-B.); milagros.delarosa@ulpgc.es (M.D.l.R.-H.); 2Hospital Insular de Gran Canaria, 35016 Las Palmas de Gran Canaria, Canary Islands, Spain; 3Department of Mathematics, Faculty of Mathematics, University of Las Palmas de Gran Canaria, 35017 Las Palmas de Gran Canaria, Canary Islands, Spain; juanjosegonzalez@ulpgc.es; 4Continuous Training Department, Primary Care Management, Gran Canaria Health Area, 35006 Las Palmas de Gran Canaria, Canary Islands, Spain; mreynohq@gobiernodecanarias.org

**Keywords:** peripheral oximetry, pulse oximeters, environmental factors, temperature, humidity, illuminance, noise, skin phototype

## Abstract

Pulse oximetry is a non-invasive, cost-effective, and generally reliable instrument measuring pulse rate and peripheral oxygen saturation (SpO_2_). However, these measurements can be affected by the patient’s internal or external factors, including the type of pulse oximeter device (POD). (1) This study’s objective was to identify potential environmental factors that may impact the measurements taken by three PODs. (2) Methods: A descriptive–analytical cross-sectional study was designed. The patients’ SpO_2_ levels were measured using a standard monitor and two PODs owned by the professionals. The measurements were taken on the patients’ fingers. Concurrently, we evaluated the surrounding environmental conditions, encompassing temperature, humidity, illuminance, and noise. (3) Results: This study involved 288 adult participants in the sample. For each 20-decibel increment in noise, there was a reduction in SpO_2_ by an average of 1%, whereas for every additional degree of ambient temperature, SpO_2_ decreased by an average of 2% (4) Conclusions: Significant correlations between SpO_2_ and age, as well as with noise and ambient temperature, were observed. No significant differences between oxygen saturation and lighting or humidity were observed. This study was prospectively registered with the Clinical Research Ethics Committee of Gran Canaria at the Dr. Negrín University Hospital, with protocol code 2019-247-1, and approved on 24 May 2019.

## 1. Introduction

Oximetry is considered the “fifth vital sign” [1,2]. It provides a simple, non-invasive, and cost-effective method of indirectly measuring oxygen saturation throughout the assessment of peripheral oxygen saturation (SpO_2_) and the estimation of oxygen blood pressure (PaO_2_). Pulse oximeter devices (PODs), or saturation meters, use spectrophotometry; by passing light through the patient’s skin, changes in the absorption of light by the blood can be measured. The devices utilize two light wavelengths: 660 nm (red) to gage the absorption of deoxygenated blood (reduced hemoglobin) and 940 nm (infrared) to measure oxygenated blood (hydroxyhemoglobin) [3,4]. SpO_2_ values greater than 95% correspond to a PaO_2_ range of 80–120 mmHg, whereas values less than 92% may indicate the need for arterial blood gas analysis [5]. Arterial blood gasses are the gold standard method of measuring arterial and venous blood oxygen levels; still, they are invasive, time-consuming, and have more economic implications.

PODs are relatively inexpensive and readily available, which has prompted healthcare professionals to utilize them in clinical practice. These devices offer an easy-to-use, practical, and simple solution that provides a wealth of pertinent clinical information for the diagnosis and treatment of patients across various medical fields. While these devices are generally robust and reliable, many lack the option for post-purchase calibration, which can potentially affect their measurement accuracy. It is currently unclear whether Spanish health institutions have an obligation to regulate the use of these devices outside of their own clinical material.

It should be noted that, in line with European Commission and FDA regulations, saturometers purchased by professionals must be specified for “medical use” in order to ensure conformity with regulatory standards [6,7,8,9]. Nevertheless, it is essential to note that previous studies [10,11] have demonstrated a high degree of concordance between different SpO_2_ measurements obtained by various PODs, even in oximeters with over three years of use and without calibration adjustments. These findings were corroborated by the Landis–Koch criteria, which yielded a “good” agreement (0.88) and a “very good” agreement (0.925) between the measurements.

The literature demonstrates how various patient-specific situations influence SpO_2_ values. These include decreased hemoglobin [12], pyrexia (core temperature reaches 40 °C, with an average SpO_2_ of 85.8%) [13], peripheral vascular disease, weak venous pulses, and dark skin, which influence the accuracy and performance of POD devices. Overestimations may occur [14,15]. Additionally, drug-bound methemoglobin, fetal hemoglobin, carboxyhemoglobin, and severe anemia [16,17,18] may also alter saturation measurements. Situations that present a risk may result in discrepancies between POD measurements, leading to variations in the alarms. This is due to the fact that some devices lack the capacity to recognize hypoxemic situations (5.4%) and bradycardia (69%) [19,20].

Furthermore, depending on the manufacturing process and the patient’s condition, PODs may exhibit varying characteristics. It must be acknowledged that the context in which the PODs are utilized, including in hospital settings, ambulances, helicopters, mountains, and other locations, is also a significant factor to be considered. In each of these scenarios, the PODs are subjected to a multitude of variables that may affect their measurements [16,17,18]. (1) The movement of the patient or transport vehicle is considered to be an artifact-inducer, representing an important source of error and false alarms [3,21,22]. (2) Interference from other devices is also a factor to be taken into account. (3) An additional consideration is the effect of altitude. In helicopters, SpO_2_ decreases from 98% at 10,000 ft to 90% at 22,000 ft [23]. Furthermore, in mountainous regions, SpO_2_ decreases from 98% at altitudes above 10,000 ft to 90% at 22,000 ft [24]. (4) Noise is another consideration [25]. (5) Intense ambient light impairs the transmission of the oximeter photodetector, resulting in a reduction in SpO_2_ [26,27]. This phenomenon is also observed in the case of light-emitting diode (LED) lamps. Blinking has been shown to negatively affect SpO_2_ due to the stroboscopic effect, with drops to 85% [28]. Under operating room lights, standard light has been found to generate an overestimation of SpO_2_ by 4% [29]. (6) Recent studies have demonstrated that an adequate ambient temperature favors precision in saturation measurements, with warm temperatures improving the quality of the signals transmitted to the device by up to four times. However, relative humidity has been shown to negatively affect this measurement [11]. (7) Environmental noise results in a 17% decrease in performance [30,31,32].

Technology has led to the development of more effective devices that can filter out potential artifacts (25% vs. 64%; *p* < 0.001) [33,34], as well as hypoperfusion and movement, thereby ensuring that SpO_2_ values remain stable (97% vs. 93%; *p* < 0.005) [34]. These devices offer advantages over their simpler counterparts.

Despite the potential for variation in SpO_2_ readings obtained by PODs, pulse oximetry is a widely used technique in various settings, offering the advantages of speed, painlessness, and cost-effectiveness. However, it is essential for health professionals to possess the requisite knowledge and familiarity with these devices to ensure accurate measurements. While some research has indicated that these factors do not necessarily affect the accuracy of SpO_2_ readings [35,36], it is nevertheless prudent to exercise caution when interpreting such results.

Pulse oximetry readings are recorded in the patient’s electronic health record (EHR) irrespective of whether the device used is staff or institutional. The origin of the device used is not indicated. The data collected in the patient’s clinical record serve as the basis for analysis, diagnosis, and medical treatment options. Consequently, it is imperative to guarantee the accuracy of the pulse oximeter (particularly if they are portable oxygen devices that dampen factors such as noise) for clinical practice. A limited number of studies have been conducted to compare the performance of low-cost pulse oximeters with those of standard medical devices. These studies have indicated that, within the 90–100% measurement range, both devices offer sufficient accuracy. However, as the measurement accuracy declines below this range, the use of low-cost pulse oximeters may be less effective [37,38].

The relevance to clinical practice of pulse oximeters (PODs) is that they are routinely utilized by healthcare personnel. However, these devices are inevitably exposed to a number of environmental factors, including illuminance or lighting levels, noise, smoke, and temperature. These environmental factors have the potential to influence the readings generated by these devices. As a consequence, the readings may exceed the established reliability range set by the manufacturer. A substantial proportion of nurses currently utilize PODs, either with or without medical certification. These instruments are designed to have a reliability level comparable to certified instruments, according to the technical specifications. However, it is crucial to acknowledge that these devices cannot be calibrated. In order to ensure patient safety, it is essential to understand how environmental factors can influence the accuracy of POD readings obtained by professionals and institutional monitors. Furthermore, institutions should be aware that these devices must meet medical use criteria for entry screening. The objective of this research is to identify the impact of the most influential environmental factors—namely, lighting, temperature, noise, and humidity—on SpO_2_ measurements recorded by three PODs on a hospital ward in Gran Canaria.

## 2. Materials and Methods

This study followed the recommendations of Strengthening the Reporting of Observational Studies in Epidemiology (STROBE) [39]. It uses the INVOLVE definition of PPI. It is reported according to the Guidance for Reporting Involvement of Patients and the Public (GRIPP 2) [40].

### 2.1. Design

This study was descriptive, cross-sectional, and analytical.

### 2.2. Background

This study was conducted over 3 months (April to June 2021) at the Orthopaedic and Traumatological Surgery Unit (HUOTS) of the Hospital Insular de Gran Canaria.

### 2.3. Participants

The participants were patients who were admitted to the HUOTS during the study period. The inclusion criteria were as follows: aged 18 years or older, not having fractured or amputated upper limbs, fingers clean and nails short, without any nail varnish, and not having permanent or temporary cognitive impairment. Patients were selected in the following way: (1) new admissions to HUOTS were checked daily. (2) EMR was checked daily. (3) Subjects were assessed for cognitive impairment and temporal disorientation. (4) Patients who met the inclusion criteria were visited in their room, presented with an information sheet, informed about data protection, and asked for informed consent. (5) Subsequently, subjects were enrolled.

### 2.4. Variables

The variables included in this study were age (years); sex (male/female); skin phototype (SPT) I—VI; peripheral oxygen saturation (percentage—%); ambient temperature (temperature in grade Celsius = °C); relative humidity (RH in percentage—%); illuminance or lighting level (lux = lx); and noise (decibel = dB).

### 2.5. Instruments

The following assessment tools were used for data collection: (1) own digital data collection template; (2) EMR; (3) a newly acquired CONTEC oximeter device (CMS50D1) (Qinhuangdao/China): CE and FDA certified, SpO_2_ measuring range 0~100%, 1% resolution for SpO_2_, measuring accuracy: ±2% at 70~100% SpO_2_ level and indeterminate at levels below 70%; (4) a CONTEC oxygen sensor (CMS50D) (Qinhuangdao/China) used for ten years without calibration: CE certified, SpO_2_ measurement range 35~99%, 1% resolution for SpO_2_, measurement accuracy: ±2% at 70~100% SpO_2_ level and indeterminate at levels below 70%; (5) one monitor with Mindray BLT Q5 (Shenzhen/China) peripheral pulse oximetry from a monitoring institution: SpO_2_ measurement range: 0~100%, resolution: 1% for SpO_2_, measuring accuracy: ±2% at 70~100% SpO_2_ level; ±3% at 40~69% SpO_2_ level and unspecified SpO_2_ level of 0~39%; (6) a Testo 540 lux meter: measuring range: 0 lx~99. 999 lx, resolution 1 lx and measurement accuracy ±3 lx; (7) a Testo 815 sound level meter: measurement range: 32 dB~130 dB, resolution 0.1 dB and measurement accuracy ±1 dB; (8) a Testo 610 thermohygrometer for measurement of temperature and RH (temperature: measurement range: −10 °C~+50 °C, resolution 0.1 °C and accuracy ±0.5 °C; RH: measurement range: 0~100%, resolution 0.1%, and measurement accuracy ±2.5%); (9) and the SPT Fitzpatrick Scale [41]: I—the skin always burns/never tans, SPT II—burns easily/tans poorly, SPT III—sometimes burns/always tans, SPT IV—never burns/tans easily, SPT V—rarely burns/tans easily and moderately pigmented, and SPT VI—rarely burns/tans promptly and heavily pigmented.

### 2.6. Procedure

Measurements were performed as follows: (1) Access to the EMR was provided to the investigator with information on compliance with inclusion criteria. (2) The room was visited to determine if the person wished to participate in the study; and information sheet and consent form were given. (3) The prototype of the skin (I–VI) was evaluated. (4) Environmental measurements were taken with the climatological meter (temperature, RH, noise and illuminance or lighting level), all at the same location, close to the patient’s hands where SpO_2_ was measured. (5) SpO_2_ measurements were taken by placing the CMS50D oximeter on the middle finger of the right hand while the CMS50D1 oximeter was placed on the middle finger of the left hand, for 15 s. The same technique was repeated three times, and then the average of the three measurements was taken. (6) Next, reverse placement (CMS50D on the left hand and CMS50D1 on the right hand, on the middle finger) was recorded. (7) The monitor oximeter was placed on the ring finger at each reading of the wireless devices (CMS50D/CMS50D1). All measurements were recorded in writing and the average of 6 measurements was calculated.

### 2.7. Statistical Methods

The median, SD, and range were used as statistical variables. The correlation between the independent variables and the measurements of the three PODs was explored by Pearson Correlation. The unit of analysis was the mean of the three measurements for each instrument. The SPSS 28.0 statistical package (Armonk, NY, USA) [42] with the appropriate license was used for all statistical analyses. The R software package (4.2.3) [43], a statistical computing environment that includes tools for data analysis and graphing, was also used. The effect of illuminance, humidity, and noise on the measurements was studied by linear regression.

## 3. Results

The sample consisted of 288 subjects. The mean age was 64.2 years [range 20–88]. Men comprised 63.9% (*n* = 184). All participants had SPT II (56.6%, *n* = 163) and SPT III (43.4%, *n* = 125).

Descriptive statistics are presented in Table 1. The mean ambient temperature was 24.2 °C. Illuminance was measured at 114.6 lx, RH at 61.0%, and noise at 54.2 dB. The proportion of participants whose beds were located close to the large window was 56.9% (*n* = 164), compared with 43.1% (*n* = 124) of the beds located close to the door. Notably, compared to the bed next to the room door (under the air conditioning vent), the bed next to the window had a higher average temperature and lighting level, as well as lower RH and average noise. It was therefore the latter variable that showed the greatest variation. Table 2 shows the descriptive analysis of the oxygen saturation measurements in the three oximeters, with the new POD presenting the greatest deviation.

The Shapiro–Wilk test (<0.0001; <0.05) revealed that the age variable did not have a normal distribution. Therefore, a non-parametric Spearman Rho test was performed (Table 3). There was a negative correlation observed between age and SpO_2_ values across the three PODs. Specifically, as age increased, SpO_2_ values decreased. This effect was particularly noticeable in the PODs owned by professionals.

The contingency coefficient (T-Test) did not reveal any significant differences (*p* > 0.05) between the sex of the participants and their SpO_2_ values (Table 4).

A bivariate Student’s *t*-test analysis was conducted to examine the relationship between skin prototypes and oxygen saturation levels in the three oximeters (Table 5). The results indicate that there were no significant differences between the two variables.

Table 6 displays the correlations between the mean SpO_2_ values of the three oximeters and the four environmental variables.

The noise variable was significant with a coefficient of –0.05 (Table 7). The values exposed suggest that for every 1% increase in dB, SpO_2_ dropped by an average of five hundred; so, for every 20 dB increase, saturation dropped by an average of 1%. The table shows some influence of noise on SpO_2_ values; however, variations in noise within a hospital room generally did not appear to be clinically relevant.

Furthermore, from the standardized coefficients, the table shows that the most influential variable on saturation was room temperature (with SpO_2_ decreasing on average by 2% for every one-degree increase in temperature), followed by age (with SpO_2_ decreasing on average by five-hundredths for every one-degree increase).

## 4. Discussion

Based on the obtained results, it was expected that age would have a negative correlation with SpO_2_ values due to anatomical–physiological changes that occur during aging. These changes include the shortening and stiffening of the rib cage, which decreases the efficiency of breathing and alters lung volumes. It results in a decrease in peak expired volume and vital capacity [44]. Nevertheless, the PODs owned by the professionals exhibit more significant values, which could be interpreted as being more sensitive or influenced by age. In terms of care activity, they provide lower SpO_2_ data than the institution’s monitor. Medical treatment may be prescribed based on these data, which may not be appropriate. No significant difference was found between sex and SPT. Previous studies have found no significant differences in the latter factor but have reported a margin of error in black skin of +3 to +5% [45].

Data revealed that the humidity and illuminance levels in the room did not affect the SpO_2_ values reported by the three oximeters evaluated in this study. The average illuminance was 114.6 lx, with a maximum value that was significantly higher than the average (1240 lx). Previous studies have demonstrated the influence of ambient light on photodetector oximeters, resulting in a reduction in SpO_2_ values [26,27]. Additionally, the association between wall luminaires in operating rooms has been investigated, with an estimated increase in SpO_2_ of 4% when compared with blood gas results [29].

Schultz et al. [46] also identified a correlation between high-intensity LED surgical ceiling luminaires, which can attain considerably higher values (100,000 lx) than the present study. Badgujar et al. [47] also observed interference from radio frequency and high-intensity light sources, including infrared, xenon, fiber optics, and fluorescent light. In addition to light, other studies have considered the effect of flickering lights on SpO_2_ measurements. This interruption has been shown to negatively affect SpO_2_ readings due to the stroboscopic effect, with drops in saturation levels of up to 85% being observed [28].

The mean humidity observed in the present study was 50.2% and did not appear to exert any influence on SpO_2_ levels. However, the bed situated adjacent to the entrance to the room exhibited the highest mean relative humidity. In contrast, another study conducted at the same hospital [11] involving a comparable mean humidity (50.85%) revealed a significant negative correlation between relative humidity and SpO_2_ levels (r = –0.321).

Noise also affected SpO_2_ measurements in our study. For every 20 dB increase, SpO_2_ decreased by an average of 1%. This variable should not be ignored in the case of temporary loud noise, although noise is usually controlled in a healthcare facility to promote a quiet environment. However, loud noises can occur in healthcare environments, as was the case when the data were collected with noise levels of up to 77 dB (mean 54.2 dB). This can interfere with measuring SpO_2_. Noise should be considered not only as an environmental factor affecting the measurements of PODs but also as an influence on the professionals performing the measurements. It has been found that auditory signals (noise) had a significant effect of 17% on the ability to detect changes in transmitted SpO_2_ concentrations [32]. In 2012, Buxton et al. [48] found noise levels of up to 80 dB in hospital rooms. This is equivalent to the noise of a vacuum cleaner or a busy street in an urban environment. Although the focus of this study was not solely on the effects of noise, it should be noted that noise is correlated with major disorders. For example, it disrupts sleep and has been associated with hypertension, cardiovascular events, immunological disorders, hormonal stress, memory deficits, and depressive states [49,50,51,52,53].

Spanish legislation (R.D. 1038/6 July 2012) [54] stipulates that noise levels in healthcare facilities should not exceed 40 dB in the morning and afternoon, and 30 dB at night. The measurements in this study were taken during the day.

During the three-month study period, the noise level remained above the established standard with a minimum of 41 dB and a maximum of 71 dB due to noise from mobile phones, televisions, music, and conversations in the corridors. In the conducted study, it was found that for every 20 dB increase in noise, there was a corresponding decrease of 1% in oxygen levels. When participants were exposed to the maximum level of 71 dB (maximum measurement), their SpO_2_ levels could decrease by up to 3.5%. This level of decrease could result in patients with mildly inadequate respiratory rates, such as those at 92%, experiencing a further drop to 88.5%, requiring blood gas measurement. Nevertheless, in addition to this, it is important that PODs can detect noise from sources other than the environment. This problem can be solved by using low-noise optical probe sensors that reduce motion interference. These sensors act as dampers and help to reduce ambient light and electrosurgical noise, as well as noise caused by patient movement.

It is noteworthy that the mean noise level was greater in the bed adjacent to the door, which suggests that the noise that was recorded emanated from inside the hospital (including sounds from staff, laundry trolleys, and meals) and exceeded that from the street outside the window, indicating the necessity to regulate the noise originating from the facility.

The most influential variable in SpO_2_ measurement was undoubtedly the ambient temperature. On average, a one-degree increase in ambient temperature reduced the SpO_2_ obtained by 2%. Further research [55] has shown that higher temperatures lead to lower SpO_2_ values, whereas hypothermia leads to a higher percentage of SpO_2_ values. This may be due to an increased venous oxygen saturation in warm environments. However, Fluck et al. [56] confirmed that natural light had little effect on operating room values. In our study, a temperature range of 23.3 to 25 °C was identified, which is within the normal range and supports the finding that SpO_2_ decreases with increasing temperature.

Another significant variable identified was age, affecting our results, with SpO_2_ decreasing by five-hundredths for every year of age. This finding is consistent with the observation by Sarabia et al. [57] that SpO_2_ decreases with increasing age. Furthermore, physiological changes that occur during the aging process cause a loss of lung tissue, particularly a reduction in alveoli, capillaries, and elasticity due to decreased elastin [58].

For other scenarios where patients have experienced changes in SpO_2_, including emergency care in helicopters, SpO_2_ has been shown to decrease from 98% at 10,000 ft to 90% and 22,000 ft [23]. Similarly, ascending to high altitudes or camping in the mountains can also result in a significant decrease in SpO_2_ levels, with a decrease of 68.0 + 9.3% at altitudes over 26,000 ft [24]. A study using multiple saturation meters would be valuable, as the literature only includes the use of a single oxygen saturation meter.

### Study Limitations

A limitation of this study was the inability to compare SpO_2_ readings from the three PODs with data collected by blood gas. Environmental noise and other factors (e.g., fatigue, stress, lack of sleep) may affect the ability of healthcare professionals to detect changes in SpO_2_. However, these variables were not included in this study [59,60]. Participants with SPT I, IV, V, and VI were not found by the researchers, which prevented assessments of the influence of melanin quantity on the SpO_2_ of the three PODs. This may be a limitation of this study, although some studies have shown a lack of association [61]. Limited access due to health facility restrictions imposed by the COVID-19 pandemic and epidemiological circumstances resulted in a delay in the data collection period to April–June 2021.

## 5. Conclusions

Our research shows that both intrinsic factors (such as age) and extrinsic factors (such as ambient temperature and noise) have a considerable impact on the accuracy of SpO_2_ measurements taken by PODs. While these devices offer the distinct advantage of professional monitoring, our study also highlights the need for careful attention to the high levels of noise present in hospital rooms as a significant variable affecting measurements in the center we studied. It is important to consider the reliability of these devices, which falls within the range of ±2% to ±4%. However, their reliability cannot be guaranteed in the presence of interfering environmental factors or operator error during the use of the device. Before this study, the researchers believed that institution-owned monitors would be less affected than practitioner-owned monitors, but it appears that noise and ambient temperature have a greater impact on the former. The use of PODs offers clinicians the advantage of being able to address respiratory pathology and monitor their patients’ hemodynamic status in the early stages. Otherwise, PODs are a cost-effective and non-invasive method of prompt data collection. However, accurate use of the device requires up-to-date knowledge of its limitations, use, and factors that may affect readings. It is interesting to mention the human influence on SpO_2_ emissions, as studies show that visual attention load significantly affects the ability to accurately detect a change in SpO_2_, which is exacerbated by the presence of auditory noise. It may affect the ability to detect fluctuations in SpO_2_ and the speed of response by the professional. These aspects would be of great interest to include in future lines of research to enable the addition of variables such as fatigue, sleep deprivation, stress, interpersonal factors, and alarm fatigue in the professionals performing the measurements. Ultimately, to ensure maximum safety for the user, it is recommended to study as many of the factors (including skin phototype) that may affect a patient’s SpO_2_ as possible_._

## Figures and Tables

**Table 1 healthcare-12-00940-t001:** Descriptive distribution of the environmental variables. *n* = 288 (R); *n* = 124 (D); *n* = 164 (W).

Variables	Mean			Std. Deviation			Minimun			Maximun		
	R	D	W	R	D	W	R	D	W	R	D	W
Temperature	24.21	24.20	24.22	0.51	0.51	0.507	23.30	23.30	23.30	25.00	25.00	24.900
Illuminance (lx)	114.67	107.10	120.39	198.79	192.76	203.63	0.00	0.00	2.00	1284.00	866.00	1284.00
Humidity (%)	54.20	54.27	54.15	4.25	1.76	5.43	21.80	51.80	21.80	54.80	57.50	59.80
Noise (dB)	54.28	54.72	53.95	9.25	8.21	9.98	41.00	41.20	41.00	77.00	77.00	71.50

R = room; D = bed next to the door; W = bed next to the window.

**Table 2 healthcare-12-00940-t002:** Descriptive analysis of the oxygen saturation measurements in the 3 models. *n* = 288.

Variables	Mean	Std. Deviation	Minimun	Maximun
Ward Monitor (calibrated)	96.61	3.25	84	100
POD-CM550D1 (new)	94.47	3.87	80	99
POD-CM550D (old)	95.87	3.18	83	99

**Table 3 healthcare-12-00940-t003:** Spearman correlations: age and SpO_2_ levels.

		Age
Mean SpO_2_ Monitor	Spearman Correlation	–0.180 **
	Sig. (2-tailed)	0.002
	N	288
Mean SpO_2_ CM550M (new)	Spearman Correlation	–0.206 ***
	Sig. (2-tailed)	<0.001
	N	288
Mean SpO_2_ CM550D (old)	Spearman Correlation	–0.218 ***
	Sig. (2-tailed)	<0.001
	N	288

** Significant at the 0.01 level. *** Significant at the 0.001 level.

**Table 4 healthcare-12-00940-t004:** Independent samples *t*-test.

POD	W	*p*	Rank-Biserial Correlation	SE Rank-Biserial Correlation
Mean SpO_2_ Monitor	10,832.000	0.059	0.132	0.071
Mean SpO_2_ CM550M (new)	10,720.000	0.087	0.120	0.071
Mean SpO_2_ CM550D (old)	9520.000	0.944	−0.005	0.071

**Table 5 healthcare-12-00940-t005:** Bivariate analysis *t*-test: STP and SpO_2_.

POD	*t*	Df	*p*
Mean SpO_2_ Monitor	–0.014	288	0.989
Mean SpO_2_ CM550M (new)	0.921	288	0.358
Mean SpO_2_ CM550D (old)	0.659	288	0.510

**Table 6 healthcare-12-00940-t006:** Correlations: environmental variables and SpO_2_.

Variables	Humidity	Temperature	Illuminance	Noise
Ward Monitor (calibrated)	0.135 *	−0.223 ***	0.045	0.066
POD-CM550D1 (new)	−0.097	−0.104	0.007	0.005
POD-CM550D (old)	−0.074	−0.055	0.092	−0.103

* *p* significant level = 0.05, *** *p* significant level = 0.001.

**Table 7 healthcare-12-00940-t007:** Model coefficients—monitor.

	Estimate	SE	*t*	*p*	Stand. Estimate
Intercept ^a^	150.22	10.29	14.598	<0.001	
Bed (door/window)	0.04	0.02	1.77	0.08	0.11
Gender	−0.32	0.43	−0.75	0.46	−0.09
Age (year)	−0.05	0.01	−3.65	<0.001 ***	−0.22
Temperature (°C)	−2.09	0.39	−5.33	<0.001 ***	−0.33
Illuminance (lx)	0.002	<0.001	1.72	0.09	0.99
Humidity (%)	<0.001	<0.001	−0.27	0.79	−0.01
Noise (Db)	−0.049	0.02	−2.29	0.02 *	−0.14

^a^ Represents reference level, * *p* significant level = 0.05, *** *p* significant level = 0.001.

## Data Availability

The data that support the findings of this study are available on reasonable request from the corresponding author, Diaz-Gonzalez C.M., on behalf of all authors.

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
