# Peer review of "Hospital Environmental Factors That Influence Peripheral Oxygen Saturation Measurements: A Cross-Sectional Study"

_healthcare, 2024, doi:10.3390/healthcare12090940_

Round 1

Reviewer 1 Report (Previous Reviewer 3)

Comments and Suggestions for Authors

I am not sure whether the comments regarding the 3rd major point were successfully addressed. In my first comment, I said the previous literature should have been introduced before discussing the results. Also, the studies should have been more specific to SpO2. However, your comment is, 'The discussion includes information on the status of environmental variables before presenting the results.'

Please reorganize the paper and also include SpO2 literature. Additionally, when addressing the comments, please include the page and line number of the exact revision.

Comments on the Quality of English Language

n/a

Author Response

Dear Reviewer,

In response to the initial peer review (four reviewers), the manuscript was updated to address the suggestions. The changes were highlighted in red. Upon further review of the document's recommendations, we have updated it again. For enhanced readability, we have highlighted key sections in blue.

LINES 46-108

LINES 285-303

Despite our best efforts, we have been unable to locate any literature specifically addressing articles on peripheral oximetry and environmental variables. Nevertheless, we have succeeded in identifying and providing data from studies that reference SpO2 measurements. Regarding your suggestion about the need to extensively improve the quality of the English language, initially it was indicated “not qualified to assess the quality of English”. We would appreciate clarification to better understand your proposal. One of the authors has a bilingual education and has completed postgraduate studies at the University of England without any difficulty.

Please find attached the manuscript with the changes in blue, to facilitate identification.

Thank you for your time in reviewing our manuscript.

Yours sincerely,

Reviewer 2 Report (Previous Reviewer 2)

Comments and Suggestions for Authors

Thanks for following the suggestions, no other things from my side.

Author Response

Dear reviewer.

The authors would like to thank you for your time and suggestions on our manuscript.

Regards.

Reviewer 3 Report (Previous Reviewer 1)

Comments and Suggestions for Authors

The authors have satisfactorily made all the corrections and the manuscript is approved.

Author Response

Dear reviewer.

The authors would like to thank you for your time and suggestions on our manuscript.

Regards.

Round 2

Reviewer 1 Report (Previous Reviewer 3)

Comments and Suggestions for Authors

Even native speakers often use editing services. I was hoping for a better logical flow, but at this point, the way the paper is written is sufficient. Thank you for your hard work.

This manuscript is a resubmission of an earlier submission. The following is a list of the peer review reports and author responses from that submission.

Round 1

Reviewer 1 Report

Comments and Suggestions for Authors

The authors present an important study on the impact of hospital environmental factors on peripheral oxygen saturation measurements.

To increase the quality of the manuscript, the authors must:

1. The SpO2 level in the patient's fingers using a standard monitor and the three PODs shows measurement error due to other factors such as skin color, nail cleanliness, finger size, etc. How were these factors excluded?

2. Important is a characterization table of the 288 participants in the study

3. It is vital to unify the units of measurement to be able to compare. For the results "For every 20 decibel increase in noise, there was a reduction in SpO2 by an average of 1 percentage point, while for each additional degree of ambient temperature, SpO2 decreased by an average of 2 units" should be unified (percentage different from units)

4. Results are missing for these factors: Age (years); Sex (male/female); Relative humidity (RH in percentage - %); and illuminance or lighting level (lux = lx). Correlated impact?

5. Several references must be updated and reviewed:

Ref. 1, 3, 4, 10-13, 16, 18, 23-25, 27, 29, 34-36, 39, 41, 42, 44-46

The ref. 35 DOES NOT apply to the format

The refs. NOT EXISTING: 48-55

Comments on the Quality of English Language

It is necessary to review the use of scientific English. Avoid conversational language

Author Response

Dear Reviewer, 1.

Thank you for your time and consideration when reading the manuscript. Please find below our responses to each of your comments:

  1. Skin colour was assessed based on photoreactivity, (only population sample with skin phototypes II and III were available). All patients had clean nails. Nail size was measured for women without nail varnish, as specified in the inclusion criteria, but finger size was not measured. Patients admitted to the surgical unit did not have nail varnish and had short nails.
  2. The authors have omitted the table due to the limited characteristics of the sample, which only includes age, gender, and skin type. Additionally, the authors are approximately 200 words short of the required 4000 words.

  1. Units have been standardised by using 'percentage' instead of other units of measurement in the abstract and throughout the text of the manuscript.
  2. A table displaying correlations between the sample variables and the SPO2 performed by the 3 PODs is included (refer to Tables 3, 4 and 5). The table shows the expected relationship between age and the decrease in SpO2 in the three oximeters, which is due to the anatomical and physiological changes in the thoracic and pulmonary cage typical of ageing.
  3. References have been checked and acknowledged.

Thank you for accepting to review our manuscript.

Yours faithfully

Reviewer 2 Report

Comments and Suggestions for Authors

I had the privilege of reviewing this interesting manuscript, detailing the effects of some environmental factors on the accuracy of pulse oxymeters. Some minor issues and suggested improvements from my side:

- 2.5 instruments: for each manufacturer also add legal headquarters/ city country etc. 

- 2.6 procedures: put all tenses to the past

- 2.7 stats: add version (for R) and proper citation for both stat programs

- 3 Results: correct all commas to points in data when indicating decimals (also somewhere in discussion); just state one of the two sexes when detailing this demographic (defining both can be redundant)

- Discussion: the authors shall take into account also prehospital emergency medicine or wilderness medicine when relating with all the described conditions and variables. Are there any studies in these fields? E.g., according to the authors' findings, pulse oxymetry could be less reliable when transferring the patients on an ambulance or (worse scenario) inside a helicopter (105 dB or more inside the cabin); or while rescuing a person on the beach or in a very humid forest? Adding some of such insight could make the manuscript tasty for the PHEM audience and prompt further studies [ and citations for the present manuscript..].

Reference 35 is not cited properly in references; also, refs at the end are probably an error of the reference manager software - please revise.

Eager to read the updated version. 

Kind regards,

Author Response

Dear Reviewer, 2.
Thank you for taking the time to read the manuscript and for your valuable feedback. Please find our responses to each of your review points below:

2.5 instruments: the jurisdiction/country has been included.
2.6 Procedures: all tenses are in the past tense.
2.7 Version (programme R) has been added and both have been quoted.
3. Results and discussion: the data now uses dots instead of commas. Only the majority sex in the sample is displayed. We have included evidence for the use of oximetry in emergency situations.

References. Removed erratum in citation 35 (DOI of former citation).
Thank you for accepting to review our manuscript.
Yours faithfully

Reviewer 3 Report

Comments and Suggestions for Authors

Major Points

1) The abstract does not adhere to the typical format of ‘Healthcare’. Please follow the 'Healthcare' format.

2) Line 77: “However, some studies have shown this not to be the case [23,24].” What do these studies say specifically? Please elaborate.

3) What did the previous literature say about the effect of hospital environment on SpO2? Specifically, temperature, illuminance, humidity, and noise? Although you briefly mention those in the discussion, it would be great if you can predict the relationship before you introduce the results. In addition, the studies mentioned in the discussion section are more general and not specifically focused on SpO2 settings.

4) What are the contributions of your study?

Minor Points

1) Lines 65-67: “In previous studies, [18,19] researchers reported a "good" and "very good" agreement (according to the Landis-Koch criteria), between the different oximeters used by various OPs. What does ‘Ops’ stand for? Please specify the acronym when you first use it.

2) It would be great if you could state in the abstract that the humidity and illuminance of the room did not affect the SpO2 values.

3) Lines 266-268: “Environmental noise and other factors (e.g., fatigue, stress, lack of sleep) may affect the ability of healthcare professionals to detect changes in SpO2. However, these variables were not included in this study [44,45].” Didn’t this study examine the effect of noise?

Comments on the Quality of English Language

Minor editing of English language required

Author Response

Dear Reviewer, 3.
Thanks for your time and consideration in the review of our manuscript.  Below we have provided you with responses to each of the points you raised in your review:
1) ABSTRACT. The text has been formatted appropriately.
2) Line 77. Clarification has been included in the paper.
3) The discussion includes information on the status of environmental variables before presenting the results.
4) Contributions of the study: To highlight the presence in the healthcare environment of oximeters purchased by professionals, many of which are not certified for "medical use" and are easily and cheaply available in online shops. These devices are used without control in health centres, and the data obtained during patient care are recorded in medical files used by doctors to determine treatment. It may jeopardize patients’ safety. Contrary to what the researchers initially stated, the vital signs monitor in the institutions were more affected by the variable noise and ambient temperature, leaving the door open for further research into the influence of the professionals' own variables (hours of sleep, stress, etc.) on the readings of changes in SpO2.
Minor points:
1) Corrected
2) Included: SpO2 values were not affected by humidity or room illuminance.
3) Line 266-268. Did the study not investigate the impact of noise?  This study looked at noise levels in two areas of the room, the patient's window and the patient's door, and their effect on SpO2 levels. However, it did not look at the noise levels experienced by the professionals and how this might lead to fatigue, stress, and affect the detection of important changes in the patient's SpO2. This could be a potential area for future research.

We are grateful for your agreement to the review of our manuscript.

Yours faithfully

Reviewer 4 Report

Comments and Suggestions for Authors

Reviewing the manuscript made it apparent that the study's objectives were challenging to discern. The title lacks scientific precision and appears disconnected from the clinical context of oxygen saturation measurement. The introduction fails to align with the study's title, predominantly focusing on the application of pulse oximeters. Consequently, the overall content does not adequately substantiate the study's aim.

An inconsistency arises when the abbreviation "OP" is used instead of the more conventional "PO" to denote the pulse oximeter. This inconsistency must be rectified for clarity and conformity with established terminology.

The methodology exhibits weaknesses that necessitate substantial improvements. While the inclusion and exclusion criteria were adequately explained, there was an oversight in not addressing fracture cases when the study was conducted in an orthopedic and trauma ward within a hospital setting. This omission raises concerns about the generalizability of the results to diverse clinical scenarios.

The study involves multiple instruments, each lacking a comprehensive description of their roles and functions. This lack of clarity is a departure from the study's title, which centers on oxygen saturation levels. The incomplete and incongruent description of data collection methods introduces potential biases and confounders, compromising the reliability of the findings.

The rationale for measuring various environmental parameters remains inadequately justified, lacking a theoretical foundation to support their inclusion. Similarly, the rationale for conducting a correlation study is not sufficiently explained. Establishing a clear theoretical framework is essential for enhancing the scientific rigor of the study.

The results section appears to be a direct extraction from computer software output, lacking proper presentation and interpretation. The discussion lacks scientific depth and requires additional clarification, particularly in establishing clinical relevance to aid patient management. The conclusion is excessively lengthy and lacks a focused synthesis of key findings, requiring refinement for conciseness and clarity. Overall, enhancing these aspects will significantly bolster the scientific merit of the manuscript.

Author Response

Dear reviewer 4

We appreciate the time and effort the reviewers have dedicated to improving our manuscript.  

We have carefully considered their feedback and made necessary changes to enhance the quality of our work.

Thank you for your valuable contribution to this process.
